# PEDF Protects Endothelial Barrier Integrity during Acute Myocardial Infarction via 67LR

**DOI:** 10.3390/ijms24032787

**Published:** 2023-02-01

**Authors:** Jingtian Liang, Qifeng Luo, Ningning Shen, Xichun Qin, Caili Jia, Zhixiang Chao, Li Zhang, Hao Qin, Xiucheng Liu, Xiaoyu Quan, Yanliang Yuan, Hao Zhang

**Affiliations:** 1Thoracic Surgery Laboratory, Xuzhou Medical University, Xuzhou 221006, China; 2Department of Thoracic Surgery, Affiliated Hospital of Xuzhou Medical University, Xuzhou 221006, China

**Keywords:** PEDF, AMI, ZO-1, endothelial cells, 67LR

## Abstract

Maintaining the integrity and protecting the stability of tight junctions in endothelial cells is a potential therapeutic strategy against myocardial ischaemia. Laminin receptors (67LR) are highly expressed on endothelial cell membranes and are associated with endothelial barrier function. Herein, we sought to demonstrate the direct effects of pigment epithelial-derived factor (PEDF) on tight junctions between endothelial cells via 67LR during acute myocardial infarction (AMI) and elucidate its underlying mechanisms. We detected that PEDF directly increased the level of the tight junction protein zonula occludens protein 1 (ZO-1) after overexpression in vitro and in vivo using Western blotting. Evans Blue/TTC staining showed that PEDF significantly reduced the size of the infarcted myocardium. Immunofluorescence and the transwell cellular experiments suggested that PEDF significantly upregulated PI3K-AKT permeability and the distribution of ZO-1 between endothelial cells under OGD conditions. Interestingly, PEDF significantly upregulated the phosphorylation levels of PI3K-AKT-mTOR under oxygen and glucose deprivation conditions but had no significant effects on the total protein expression. The protective effect of PEDF on ZO-1 was significantly inhibited following the inhibition of PI3K-AKT-mTOR. The activation of phosphorylation of PI3K-AKT-mTOR by PEDF was blocked after silencing 67LR, as were the protective effects of PEDF on ZO-1. Therefore, we have reason to believe that PEDF increased ZO-1 expression through the 67LR-dependent PI3K-AKT-mTOR signaling pathway, thus maintaining tight junction stability and protecting cardiac function.

## 1. Introduction

Acute myocardial infarction (AMI), a cardiovascular disease, is associated with high morbidity and mortality worldwide [1,2]. Ischemia and hypoxia caused by AMI destroy endothelial connections, resulting in the loss of microvascular integrity. Endothelial connection rupture and microvascular disintegration aggravate the swelling and inflammation of surrounding tissues, along with the myocardial tissue injury and seriously affect the prognosis of these patients [3,4]. The guidelines of The American College of Cardiology/American Heart Association recommend that minimizing microvascular injury is the priority in the treatment of AMI. Therefore, it is crucial to develop treatment strategies and predict the prognosis of AMI to prevent a series of pathological changes in microvessels and reduce the rupture of endothelial cell junctions [5].

Tight junctions are widely found between epithelial or endothelial cells and are important for maintaining vascular permeability. ZO-1 is the main signaling protein controlling the integrity of the endothelial barrier and its molecular weight is 220~225 kDa. The disruption of its integrity damages the vascular endothelial barrier, increases vascular permeability and aggravates tissue damage [6,7]. The laminin receptor (67LR) is a multifunctional protein that is closely related to many pathological processes and is highly expressed in the endothelial cell membrane; however, its expression in stable blood vessels and endothelial cells is low. Our previous results showed that the phosphorylation level of LR can affect microvascular structure and function [8]. Pigment epithelium-derived factor (PEDF) is an endogenous protein of the SERPIN superfamily, which is widely expressed in several tissues such as the liver, kidney, heart, retina, and adipocytes, and exerts a variety of biological effects including anti-vascular leakage and protection of the endothelial barrier [9]. PEDF is one of the strongest known inhibitors of angiogenesis, but interestingly, PEDF targets immature neovascularization while protecting mature vessels. Accumulating evidence suggests that PEDF can act as an endogenous vascular stabilizing factor in the damaged microenvironment by interacting with receptors on the cell surface to exert its specific function and reduce pathological damage [10,11]. The laminin receptor (67LR) is a key receptor in the PEDF binding site; however, whether PEDF regulates ZO-1 expression and distribution through 67LR and subsequently affects endothelial barrier integrity during AMI remains largely unknown [12,13].

Therefore, the objectives of this study were to investigate (1) whether PEDF regulated the integrity of the vascular endothelial barrier during AMI and (2) whether PEDF regulated the expression and distribution of ZO-1 through 67LR and elucidate its underlying mechanism.

## 2. Results

### 2.1. PEDF Reduces Myocardial Infarct Area and Vascular Endothelial Permeability in AMI Rats

Our previous study showed that PEDF expression was downregulated in the infarcted area after AMI (Appendix A) [1,2]. To investigate the potential effect of PEDF on the endothelial barrier of ischemic myocardium, we used LVs to mediate higher or lower expression of PEDF in rat local myocardial tissue. The expression level of PEDF was detected by WB (Appendix A). The efficiency of viral transduction was further confirmed by immunofluorescence observation of GFP expression in the myocardial tissue (Appendix A). Subsequently, the AMI model was established by ligation of the left anterior descending branch of the rat heart. The myocardial infarct size was observed by Evans Blue/TTC staining, and the expression of cTn-I, a marker of myocardial injury, was detected to evaluate the degree of myocardial tissue injury during AMI. As shown in Figure 1A,B, AMI caused obvious infarct areas in myocardial tissues, and overexpression of PEDF significantly reduced infarct size, while its knockdown further increased infarct size. Accordingly, ELISA results showed that PEDF significantly decreased serum cTn-I concentration in AMI rats (Figure 1C). These results suggested that PEDF alleviated myocardial injury in AMI. To further assess the effects of PEDF on TJs between vascular endothelial cells during AMI, we first detected the expression of ZO-1, a hallmark protein of vascular TJ function, to characterize the vascular junction structure (Figure 1D,E). After 6 h of ischemia, the expression of ZO-1 in rat myocardial tissue decreased significantly. Compared to the AMI-Ctrl group, the expression of ZO-1 in the AMI-PEDF group increased significantly; in contrast, the expression of ZO-1 in the AMI-shPEDF group was the lowest. Subsequently, the degree of vascular leakage was detected by immunofluorescence staining for fibronectin. A significant leakage of the vasculature of myocardial tissue after 6 h of ischemia was found, which was partially reversed by the overexpression of PEDF (Figure 1F,G). These results indicated that PEDF played an active role in reducing the size of the myocardial infarct and vascular endothelial permeability after AMI in rats.

### 2.2. PEDF Maintains the Stability of Vascular Endothelial TJs under OGD Conditions

After verifying the protective effect of PEDF on the endothelial barrier in vivo, we next performed in vitro experiments. The OGD model was constructed to simulate the AMI environment in HUVECs [14]. Consistent with the in vivo results, WB showed that the OGD environment led to the degradation of ZO-1, while PEDF effectively increased the expression of ZO-1. Notably, ZO-1 expression was the lowest in the shPEDF group (Figure 2A,B). The results of immunofluorescence staining showed that the protein distribution of ZO-1 in the normal group was neat, the cell-to-cell connection was tight, and the cell structure was intact. Under OGD conditions, ZO-1 distribution on the cell membrane was disordered, and significant endocytosis was observed in the cytoplasm, which was effectively prevented by PEDF (Figure 2C). A similar phenomenon was observed when recombinant human PEDF protein was added (Figure 2D,F). Similarly, the results of the transwell experiments showed that PEDF effectively prevented OGD-induced increases in endothelial cell permeability (Figure 2G,H). It was concluded that PEDF played an active role in stabilizing the TJs of endothelial cells and reducing vascular permeability after myocardial infarction in rats.

### 2.3. PEDF Regulates the Expression of ZO-1 in Endothelial Cells through the PI3K-AKT-mTOR Pathway

Some studies have shown that the abnormal increase in the phosphorylation levels of the PI3K-AKT-mTOR pathway plays an important role in myocardial ischemic injury [15]. The PI3K-AKT-mTOR pathway is an intracellular signal transduction pathway promoting metabolism, proliferation, cell survival, growth, and angiogenesis in response to extracellular signals [16,17]. The level of PI3K-AKT-mTOR phosphorylation decreased during myocardial ischemia. To assess whether the TJ stability regulated by PEDF was related to this pathway, we performed WB to detect the expression of each related protein under OGD conditions. WB results showed that the phosphorylation levels of PI3K, AKT, and mTOR decreased significantly under OGD conditions. PEDF increased the expressions of p-PI3K, p-AKT, and p-mTOR, but the total protein expressions of PI3K, AKT, and mTOR remained unchanged. Wortmannin attenuated the phosphorylation of PI3K, AKT, and mTOR. 3CAI attenuated the phosphorylation of AKT and mTOR but showed no significant effect on PI3K. Torin2 only inhibited the phosphorylation of mTOR but did not affect the phosphorylation of PI3K/AKT. These results indicated that PEDF could increase the expression of ZO-1 through the activation of the PI3K-AKT-mTOR pathway under OGD conditions (Figure 3A–E).

### 2.4. PEDF Maintains the Stability of Vascular Endothelial TJs by Activating the PI3K-AKT-mTOR Pathway via 67LR

To elucidate the receptor involved in mediating the effect of PEDF on the TJS protein, ZO-1, during OGD, we used LVs to mediate lower expressions of 67LR and PEDFR in HUVECs. The results showed that 67LR and not PEDFR was involved in this process (Figure 4A–E). 

Immunofluorescence results also showed the decisive role of 67LR in PEDF-mediated stabilization of the distribution of the TJS protein, ZO-1, in endothelial cells (Figure 4F). These results indicated that PEDF maintained the TJs of endothelial cells through its receptor, 67LR, under OGD conditions. In addition, compared to the control group, the expressions of p-PI3K/p-Akt/p-mTOR decreased significantly in the s67LR group, along with that of ZO-1 (Figure 5A–E). In conclusion, the effect of PEDF on ZO-1 via the PI3K-AKT-mTOR pathway was mediated by its receptor 67LR.

## 3. Discussion

Herein, we describe a novel role for PEDF as a positive regulator that stabilizes endothelial TJs and protects cardiac functions against AMI. We also tested and verified that the effect of PEDF on TJs occurred through the 67LR receptor-dependent PI3K-AKT-mTOR pathway (Figure 6).

Previous studies have shown that AMI leads to the destruction of the TJs of cardiomyocytes, increases interendothelial space and permeability, and stimulates the cascade of inflammatory effects, thus aggravating the extensive damage to myocardial tissue and endothelial cells and seriously affecting the prognoses of patients with AMI [18]. PEDF was originally described as a neurotrophin but has now been recognized as one of the most potent endogenous antiangiogenic factors. It can inhibit the production of new blood vessels effectively [19,20]. The anti-tumor and anti-inflammatory effects of PEDF have been gradually confirmed [21,22]. Our study showed that PEDF could increase ZO-1 expression and promote its membrane distribution during AMI, thereby reducing endothelial cell permeability. More importantly, we observed a general decrease in the myocardial infarct size and vascular leakage in rats, further demonstrating the role of PEDF in improving the ischemic hypoxic microenvironment of AMI and maintaining interendothelial junctions. AMI involves the activation of various signaling pathways, among which the reduction of PI3K-Akt-mTOR phosphorylation is a typical feature. The mTOR signaling pathway plays an important role during myocardial infarction and it regulates cellular function in a variety of ways. Inhibiting mTOR signaling was cardioprotective during myocardial infarction in rats [23]. In addition, activation or restoration of PI3K-AKT-mTOR phosphorylation was proven as an effective means to reduce AMI injury [24]. PEDF could significantly increase the phosphorylation levels of PI3K/Akt/mTOR in AMI, and the use of corresponding inhibitors, namely wortmannin, 3CAI, and Torin2, blocked the therapeutic effects of PEDF. These results suggested that activation of the PI3K-AKT-mTOR pathway was an important mechanism by which PEDF protected the endothelial barrier.

PEDF has several membrane receptors. For a long time, several experiments focused on assessing the various biological functions of PEDF along with the different receptors [25,26]. Two transmembrane proteins have been identified as important receptors of PEDF—PEDF-R and 67LR [19]. 67LR is highly expressed in endothelial cells and is an important functional receptor of PEDF for binding to endothelial cells [27]. Our previous study showed that LR was closely related to the morphology and function of microvessels; LR could reduce the no-reflow area through stabilization of the LR-TIMAP/PP1c complex [8]. Both 67LR and PEDF-R are expressed in endothelial cells [28]. To explore the specific mechanism of PEDF action, we separately interfered with the expression of the above two receptors. The results showed that knocking down 67LR significantly blocked the biological effects of PEDF but knocking down PEDF-R had no significant effect. Thus, we reasonably speculated that the protective effects of PEDF on the endothelial barrier were mainly achieved by binding to 67LR.

PEDF, when overexpressed after experimentally inducing ischemic injury, represents a potential therapeutic modality to protect the myocardium and improve functional recovery [29]. However, there are significant limitations to the administration of intramuscular transfected PEDF. The window of opportunity for the use of PEDF may be more appropriate for “planned” ischemic events, such as after coronary arterial bypass surgery (CABG). Therefore, further studies are needed to identify the key components mediating the cardioprotective functions of PEDF. Maintenance of the stability of TJs between endothelial cells plays a key role in alleviating myocardial infarction. PEDF overexpression could alleviate myocardial infarction injury in rat hearts. Although other mechanisms may be involved, these findings still have potential translational implications for the treatment of acute coronary syndromes.

## 4. Materials and Methods

### 4.1. Animals

Sprague-Dawley (SD) male rats (250 ± 20 g, 8–10 weeks) were obtained from the Experimental Animal Centre of Xuzhou Medical College. These rats were housed with a 12-h light–dark cycle and free access to food and water. All of the animals used in this experiment were randomly divided into five groups: Sham, AMI, AMI-vector, AMI-PEDF, and AMI-sh-PEDF. Rat care and experimental protocols were approved by the Experimental Animal Center of Xuzhou Medical University (201802W007) and performed according to the National Institutes of Health (NIH) Guide for the Care and Use of Laboratory Animals.

### 4.2. Reagents and Antibodies

The BCA protein concentration assay kit was purchased from Beyotime (Shanghai, China). The tissue or cell total protein extraction kit was purchased from Biotechnology (Shanghai, China). The cTnI ELISA kit was purchased from Shanghai Renjie Biotechnology Co., Ltd. (Shanghai, China). The following antibodies were purchased: anti-rabbit serpinf1 (PEDF) antibody (Catalog No. DF6547, Affinity Biosciences, Changzhou, China); anti-goat ZO-1 (Catalog No. Ab190085); anti-mouse fibronectin (Catalog No.66042-1-Ig), anti-rabbit CD31 (Catalog No. Ab182981); anti-rabbit PI3K (Catalog No. Cell signaling 4292); anti-rabbit P-PI3K (Catalog No. Cell signaling 4228); anti-rabbit AKT (Catalog No. Cell signaling 9272); anti-rabbit P-AKT (Catalog No. Cell signaling 9275); anti-rabbit mTor (Catalog No. Cell signaling 2983); anti-rabbit p-mTor (Catalog No. Cell signaling 5536); anti-rabbit 67LR (Catalog No. Ab133645); anti-rabbit PEDFR (Catalog No. Ab207799); anti-rabbit β-Tubulin (Catalog No. 10094-1-AP); wortmannin (Catalog No. Cell signaling 9951); 3CAI (Catalog No. B4688), and Torin2 (Catalog No. B1640); DAPI staining solution was purchased from KeyGEN Biotech (Nanjing, China).

### 4.3. Lentivirus (LV) Preparation

Recombinant LVs (PEDF-LV and PEDF-shRNA-LV) were prepared by Shanghai Jikai Gene Medical Technology Co., Ltd. Briefly, PEDF overexpression plasmids and the RNAi vector were successfully constructed, packaged, and transfected in 293T cells. The concentrated titers of the LV suspensions were 2 × 10^12^ TU/L.

### 4.4. Immunofluorescence Assay

Experimental cells were seeded in 24-well plates, allowed to attain a confluency of 80–90%, rinsed, and fixed in 4% paraformaldehyde for 15 min at room temperature. After washing off the residual paraformaldehyde, 10% goat serum (0.3% Triton dilution) was added to seal the plate for 1.5 h. The cells were incubated with the primary antibody (1:200 dilution) overnight at 4 °C and on the next day, after rinsing, the samples were incubated with the secondary antibody for 2 h. After washing off the secondary antibody, the DAPI stain was added to stain the nuclei of the cells for 10 min. After washing off excess DAPI, a fluorescent anti-quenching agent was added to seal the slides. Laser confocal observation was performed and several randomly selected high-magnification fields (600×) were photographed.

### 4.5. Cell Culture and Treatment

HUVECs were cultured in the ECM supplemented with 10% fetal bovine serum in a humidified atmosphere containing 5% CO_2_. To establish the OGD cell model, sugar-free DMEM without fetal bovine serum was used to replace the complete ECM and cultured in a three-gas incubator (HEARcell 150i, 37 °C, 5% CO_2_, 1% O_2_, and 94% N_2_) for 6 h. Gene delivery to LVs was based on the cell number (infection multiplicity = 20). At the indicated time points, total protein was extracted according to the manufacturer’s instructions.

### 4.6. Western Blot Analysis

Using the Total Cell or Tissue Protein Extraction Kit, proteins were extracted from HUVECs or heart tissues. The BCA Protein Concentration Assay Kit was used to determine the protein concentration of each sample. Proteins were separated by SDS-PAGE and transferred to nitrocellulose membranes. After 15 min of blocking in rapid closure solution, the membranes were incubated overnight at 4 °C with primary antibodies (ZO-1/PI3K/P-PI3K/AKT/P-AKT/mTor/p-mTor/67LR/PEDFR/β-Tubulin, Section 4.2) against the target proteins. After washing, the blots were incubated with fluorescently-labelled anti-mouse, anti-rabbit, or anti-goat secondary antibodies for 1–2 h at room temperature and imaged using a dual infrared laser imaging system (Odyssey CLX). Densitometric analysis of bands was carried out using the ImageJ software (National Institutes of Health, Bethesda, MD, USA). Protein levels were calculated as the corresponding protein/β-tubulin expression.

### 4.7. AMI Model Generation

The tissues of SD rats were taken and fixed after intraperitoneal anaesthesia with 2% sodium pentobarbital (2 mL/kg). Using a 16-gauge polyethylene catheter connected to a ventilator (Medical Equipment Company, Shanghai, China). Routine skin preparation, disinfection, and spreading of the sterile hole tissue were performed; about 0.5 cm to the left of the median sternum, the third and fourth intercostal spaces were prodded, and the pericardium was cut open to expose the heart and explore the arterial cone. Blocking of the left anterior descending branch of the coronary artery was achieved by suturing the ligature below the arterial cone with a 6-0 non-invasive suture. After treatment, the animals were sutured back and sent to the Animal Centre for routine husbandry. Penicillin (100 mg/kg) was injected into the postoperative muscle to prevent infection.

### 4.8. Transwell Assay

HUVECs were inoculated onto microporous membranes in the upper chamber of a Transwell (0.4 μm pore size, 12 mm diameter, Corning, NY, USA) at a density of 1 × 10^5^/cm^2^, and 150 μL of ECM medium was added to the upper chamber, while 500 μL of the ECM was added to the lower chamber. The media was changed the next day. After the complete fusion of the cells, three 50 μL volumes of FITC-dextran were added to the upper chamber of each group at a final concentration of 10 μM. The normal group was then incubated in a normoxic incubator, while the hypoxic group was switched to the sugar-free DMEM and incubated in a hypoxic incubator (94% N_2_, 5% CO_2_, and 1% O_2_). The fluorescence intensity was measured using a fluorescent enzyme marker (Biotek, Vermont, VT, USA) with an excitation wavelength of 530 nm and an emission wavelength of 590 nm. The permeability was calculated based on the relative fluorescence intensity of the normal group.

### 4.9. Evans Blue/TTC Staining

The hearts were removed from the mice; residual blood was rinsed, and animals were perfused with Evans Blue, and snap frozen at −20 °C for 20 min. Then, the frozen heart was sliced at 2 mm intervals and cut into 5–6 slices. The slices were placed in 1.5% TTC, wrapped in tin foil, and incubated at 37 °C for 15 min. After washing, the sections were fixed in 4% paraformaldehyde and photographed using a stereomicroscope. The ratio of myocardial infarct area to left ventricular myocardium was analyzed and counted using Image Pro Plus software.

### 4.10. Enzyme-Linked Immune Sorbent Assay (ELISA)

All solutions were prepared according to the instructions. Specimens and standards of different concentrations were added to the corresponding wells and then the en-zyme-labeled antibody working solution was added. The wells were incubated for 3 h at room temperature and sealed; then, the plate was washed 4 times and color developer added, and incubated for 20–25 min at room temperature in the absence of light. Termination solutions were added, well mixed and the OD450 value was measured immediately.

### 4.11. Statistical Analysis

Data are expressed as mean ± standard deviation (SD). Multiple group comparisons were performed by one-way ANOVA followed by the least significant difference *t*-test for post hoc analysis. Data between two independent groups were compared using a two-tailed Student’s *t*-test. All analyses were performed using SPSS 25 software (Chicago, IL, USA). Differences with *p* < 0.05 were considered statistically significant.

## 5. Conclusions

Overall, our results suggested that PEDF binding 67LR was a protective factor of the endothelial barrier during AMI. Application of PEDF during AMI is a feasible strategy to reduce cardiomyocyte apoptosis and endothelial barrier damage. These findings provide a promising prospective approach to improving management and prognostic strategies for AMI.

## Figures and Tables

**Figure 1 ijms-24-02787-f001:**
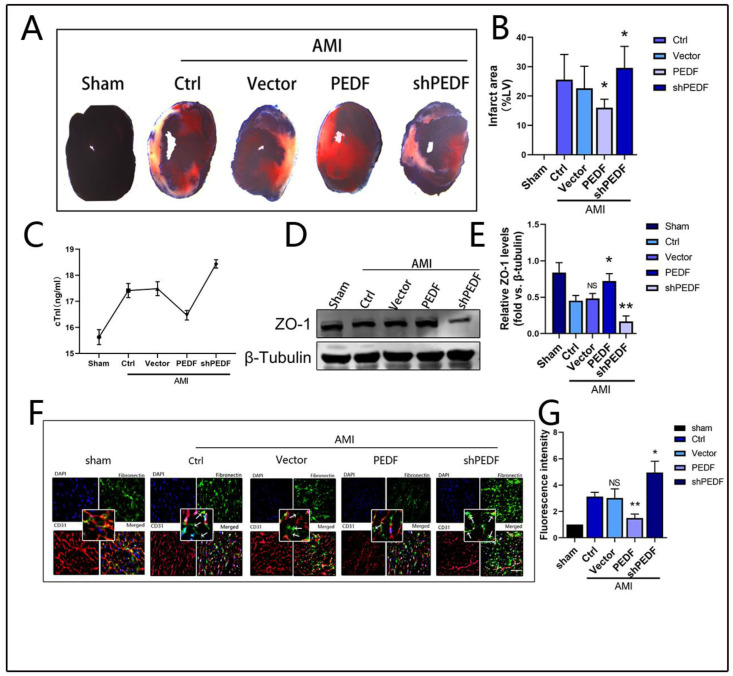
PEDF reduces myocardial infarct size and vascular leakage in AMI rats. (**A**) Representative images of the Evans Blue/TTC staining in myocardial tissues. (**B**) Quantification of myocardial infarct size. (**C**) Quantification of serum cTn-I concentration in rats. (**D**,**E**) Western blotting to detect the effect of PEDF on the expression of ZO-1. (**F**,**G**) Immunofluorescence to observe the effect of PEDF on vascular leakage; 15–20 images from each group were used for fluorescence intensity analysis. White arrows point to leakages; Scale Bar = 100 μm. Data are expressed as mean ± SD; *n* = 4; NS, no significant difference; * *p* < 0.05; ** *p* < 0.01.

**Figure 2 ijms-24-02787-f002:**
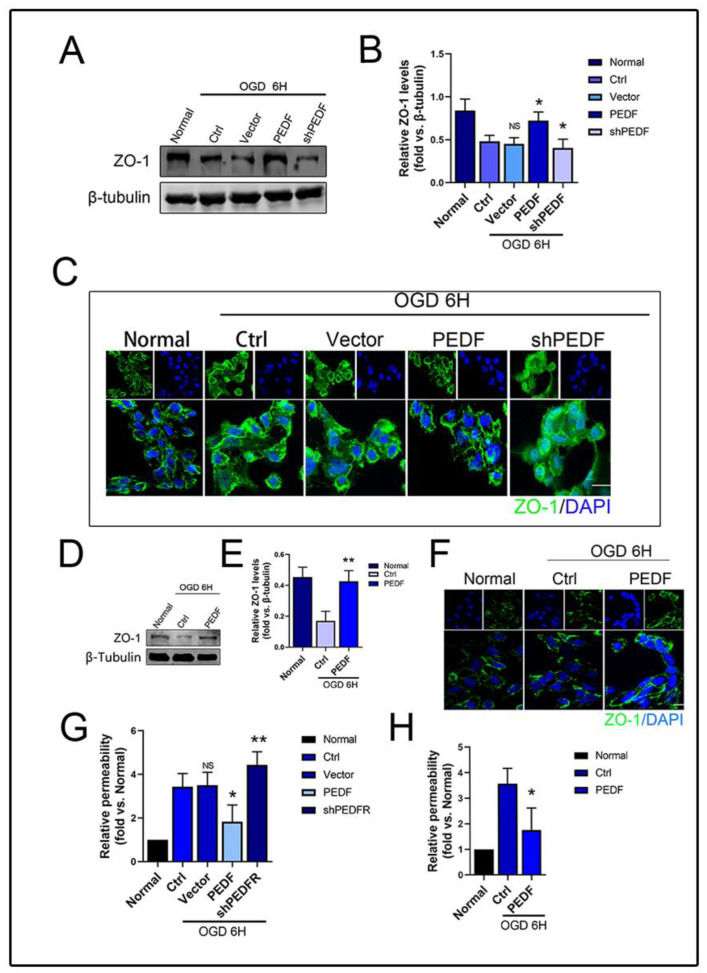
The effect of PEDF on the permeability of HUVECs during OGD. (**A**,**B**) Western blotting to detect the effect of PEDF on the expression of ZO-1 in vivo. (**C**) Immunofluorescence to observe the effect of PEDF on the distribution of ZO-1 in vivo. Scale Bar = 200 μm. (**D**,**E**) Western blotting to detect the effect of PEDF on the expression of ZO-1 in vitro. (**F**) Immunofluorescence to observe the effect of PEDF on the distribution of ZO-1 in vitro. Scale Bar = 200 μm (**G**,**H**) Quantification of the effect of PEDF on endothelial cell permeability. Data are expressed as mean ± SD; *n* = 4; NS, no significant difference; * *p* < 0.05; ** *p* < 0.01.

**Figure 3 ijms-24-02787-f003:**
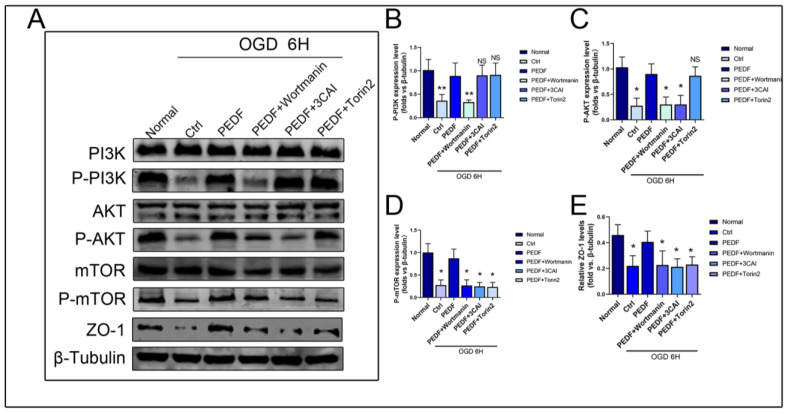
PEDF regulates the PI3K-AKT-mTOR pathway in endothelial cells during OGD. (**A**) Western blot to detect the effect of PEDF on phospho-PI3K, phospho-AKT and phospho-mTOR expressions. (**B**–**E**) Quantification of related protein expressions. Data are expressed as mean ± SD; *n* = 4; NS, no significant difference; * *p* < 0.05; ** *p* < 0.01.

**Figure 4 ijms-24-02787-f004:**
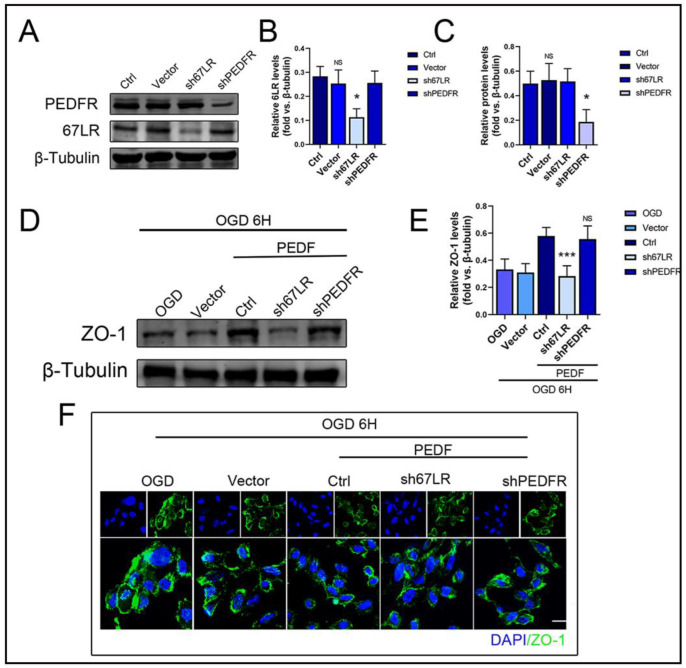
PEDF maintains endothelial tight junction stability through 67LR. (**A**–**C**) Transfection with both 67LR and PEDFR viruses was successful. (**D**,**E**) Western blotting to detect the effect of PEDF on the expression of ZO-1 in vitro; (**F**) Immunofluorescence to observe the effect of PEDF on the distribution of ZO-1 in vitro. Scale Bar = 200 μm. Data are expressed as mean ± SD; *n* = 6; NS, no significant difference; * *p* < 0.05; *** *p* < 0.001.

**Figure 5 ijms-24-02787-f005:**
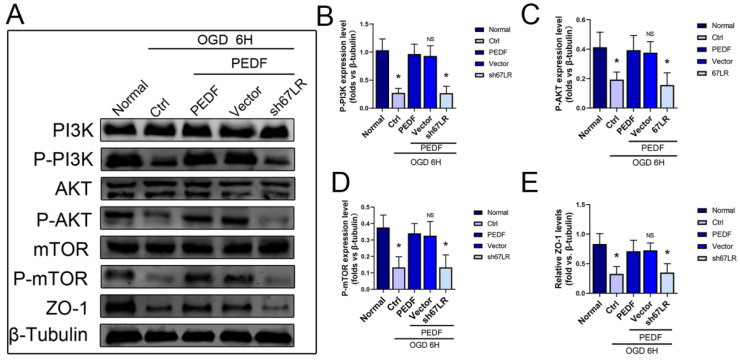
PEDF regulates the PI3K-AKT-mTOR pathway and ZO-1 expression through 67LR. (**A**) Western blotting to detect the effect of PEDF on phospho-PI3K, phospho-AKT, phospho-mTOR, and ZO-1 expressions. (**B**–**E**) Quantification of the related protein expressions. Data are expressed as mean ± SD; *n* = 4; NS, no significant difference; * *p* < 0.05.

**Figure 6 ijms-24-02787-f006:**
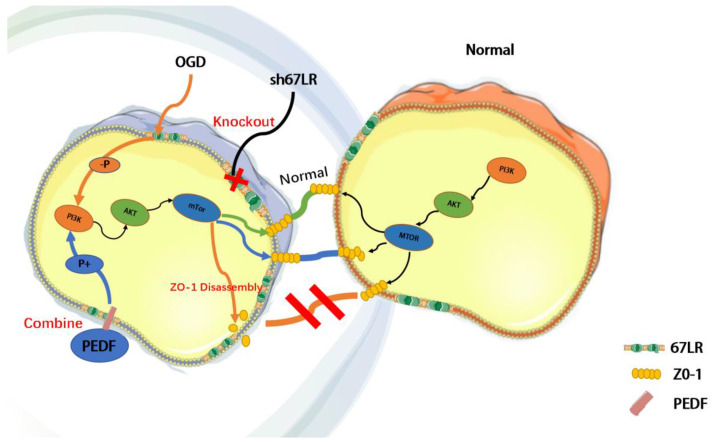
Schematic diagram demonstrating that PEDF maintains the stability of TJs by activating the PI3K-AKT-mTOR pathway via the 67LR receptor under OGD conditions.

## Data Availability

Not applicable.

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
