# Peer review of "PEDF Protects Endothelial Barrier Integrity during Acute Myocardial Infarction via 67LR"

_ijms, 2023, doi:10.3390/ijms24032787_

Round 1

Reviewer 1 Report

The manuscript by Liang et al. entitled, “PEDF Protects Endothelial Barrier Integrity during Acute Myocardial Infarction via 67LR” is aimed to demonstrate the direct effects of PEDF on tight junctions between endothelial cells via 67LR during AMI and elucidate its underlying mechanisms using a proper methodologies. The most obvious finding from this study is that PEDF was successfully prevent the degradation or destruction of the cells. It is believed that PEDF increase ZO-1 expression and promote its membrane distribution during AMI, thereby reducing endothelial cell permeability. The authors had successfully proved a novel role for PEDF as a positive regulator that stabilizes endothelial TJs and protects cardiac functions against AMI. Furthermore, the author also verified that the effect of PEDF on TJs occurred through the receptor 67LR-dependent PI3K-AKT-mTOR pathway. In future research, the author also proposed an additional study to identify the key components mediating the cardioprotective functions of PEDF.

In general, the abstract is easy to understand, however, it lacks a brief methodology. The introduction section lays out the study's rationale and the aims well explained. The data in the result section is well organized. The author describes the findings and compares them to previous ones in the literature. All the references are carefully selected and appropriate.

Overall, the manuscript is well-written, well-structured, and highly novel. The general readers will be interested in the summaries and findings. Publication would benefit the scientific community since these findings have potential translational implications for the treatment of acute coronary syndromes.

Some minor issues to improve:

Abstract

Abbreviations should be avoided in an abstract unless a term is used multiple times.

Please include a brief description of the methodology in the abstract.

Graphical Abstract

I would recommend the authors design a Graphical Abstract for this study in order to better show the whole story in a simple and informative manner.

Introduction

Line 43- Suggest including the molecular weight of ZO-1.

Methodologies

Line 71-72: In section 2.1, Please provide the code number for ethical approval as well as the date of approval.

The author should mention in section 2.1 that all of the animals used in this experiment were randomly divided into four groups: sham, AMI-control, AMI-PEDF, and AMI sh-PEDF.

Why was the 293T cell line used in this study?

Line 100- magnification field (x600) should be written as (600x).

Line 110- Please provide the manufacturer and country of the usage kit.

Line 114- Please specify the primary antibody used in the WB.

Line 122- Something is missing in this phrase. ‘A ventilator device for animals'.

Line 147-148- Please describe the cTnI concentration determination procedure.

How do you calculate the size of an infarct? I would suggest including some information about quantifying myocardial infarct size in this section.

Discussion

Line 285: Please include citations in this sentence. "For a long time, several experiments focused 284 on assessing the various biological functions of PEDF along with the different receptors [citations]".

Reference

Lu et al. (2016) is missing in the text.

Figure

Figure 1F: I would recommend enlarging the immunofluorescent image. Use an arrow to indicate vascular leakage.

Author Response

Response to Reviewer 1 Comments

We gratefully thank all reviewers for their time spend making their constructive remarks and useful suggestions, which has significantly raised the quality of the manuscript and has enable us to improve the manuscript. Each suggested revision and comment brought forward by the reviewers were accurately incorporated and considered. We have studied reviewer’s comments carefully and have made revision which marked in highlight in the paper. The comments of the reviewers are responded point by point and the revisions are indicated as following.

Reviewer point #1: Abstract 

Abbreviations should be avoided in an abstract unless a term is used multiple times.

Please include a brief description of the methodology in the abstract.

Author response #1:Thank you very much for this valuable suggestion. We have rewritten the abstract according to your suggestion. These revisions could be found in Section Abstract.

Reviewer point #2: Graphical Abstract

I would recommend the authors design a Graphical Abstract for this study in order to better show the whole story in a simple and informative manner.

Author response #2: Thank you for your suggestion. We have designed a Graphical Abstract based on research and biological mechanisms (that is Figure6). We have uploaded it as the Graphical Abstract in the submission system.

Reviewer point #3: Introduction

Line 43- Suggest including the molecular weight of ZO-1.

Author response #3: Thank you for your comment. We have added additional information in the Introduction which is the molecular weight of ZO-1. This revision could be found in Line 47-48.

Reviewer point #4: Methodologies

Line 71-72: In section 2.1, Please provide the code number for ethical approval as well as the date of approval.

Author response: Thank you very much for this suggestion. Rat care and experimental protocols were approved by the Experimental Animal Center of Xuzhou Medical University (201802W007) and performed according to National Institutes of Health (NIH) Guide for the Care and Use of Laboratory Animals. This revision could be found in Line 74-77.

The author should mention in section 2.1 that all of the animals used in this experiment were randomly divided into four groups: sham, AMI-control, AMI-PEDF, and AMI sh-PEDF.

Author response: Thank you very much for this professional suggestion. We have added the missing information according to your suggestions. This revision could be found in Line 72-74.

Why was the 293T cell line used in this study?

Author response: Thank you for this question. 293T cells are lentivirus packaging tool cells, we used 293T cells to produce recombinant lentivirus (PEDF-LV and PEDF-shRNA-LV). Reference: PMID: 35459189.

Line 100- magnification field (x600) should be written as (600x).

Line 110- Please provide the manufacturer and country of the usage kit.

Line 114- Please specify the primary antibody used in the WB.

Line 122- Something is missing in this phrase. ‘A ventilator device for animals'.

Line 147-148- Please describe the cTnI concentration determination procedure.

Author response: Thank you very much for these professional questions and valuable suggestions. The above issues have been revised one by one according to your suggestions. This revision could be found in Line 108, Line 114, Line 123, Line 131-132, Line 159-164.

How do you calculate the size of an infarct? I would suggest including some information about quantifying myocardial infarct size in this section.

Author response: Thank you very much for this question. Infarct ratio is the ratio of myocardial infarct size to the left ventricular myocardia and was analyzed and counted using Image Pro Plus software. We have added this information in Section 2.9.

Reviewer point #5: Discussion

Line 285: Please include citations in this sentence. "For a long time, several experiments focused 284 on assessing the various biological functions of PEDF along with the different receptors [citations]".

Author response #5: Thank you very much for your suggestions, we have modified it according to your suggestion. This revision could be found in Line 304-306.

Reviewer point #6: Reference

Lu et al. (2016) is missing in the text.

Author response #6: Thank you for your suggestion and we apologize for our negligence, we have added the missing information.

Reviewer point #7: Figure

Figure 1F: I would recommend enlarging the immunofluorescent image. Use an arrow to indicate vascular leakage.

Author response #7: Thank you very much for your review, we uploaded more representative fluorescent images and used arrows to indicate the leak area. Hope this could solve the issue.

Again, thanks for pointing out the insufficiency of the interpretation.

Reviewer 2 Report

Liang et al. showed that PEDF is protective against AMI through 67LR-dependent PI3K-Akt-mTOR activation. Overall, the studies are well-presented and provide information for AMI treating strategies. A few points need to be addressed to support the findings.

Figure 1F: Ischemia does not show significant difference in the shPEDF group, compared to the control or vector. Did the author quantify by fluorescence? The zoom-in square does not seem to represent the whole picture.

Figure 4D: In lane shPEDFR, ZO-1 expression level relative to Tubulin looks lower than ctrl. The author should repeat the western blot to strengthen their conclusion.

It is clear in figure 5F that, 67LR is necessary for PEDF protective function, while it’s hard to convince me that PEDFR is not involved. Adding cell membrane marker and re-analyzing data will be appreciated.

Gao et al. reports that (Gao et al., Int J Mol Med, 2020) inhibiting mTOR signaling is cardioprotective during myocardial infarction in rats. A discussion on how mTOR responses to distinct stimuli and regulates cellular functions via multiple mechanisms would improve the significance of the study. 

Supplementary figure 2 is not referred correctly in the text.  

Author Response

Response to Reviewer 2 Comments

We gratefully thank all reviewers for their time spend making their constructive remarks and useful suggestions, which has significantly raised the quality of the manuscript and has enable us to improve the manuscript. Each suggested revision and comment brought forward by the reviewers were accurately incorporated and considered. We have studied reviewer’s comments carefully and have made revision which marked in highlight in the paper. The comments of the reviewers are responded point by point and the revisions are indicated as following.

Reviewer point #1: Figure 1F: Ischemia does not show significant difference in the shPEDF group, compared to the control or vector. Did the author quantify by fluorescence? The zoom-in square does not seem to represent the whole picture.

Author response #1: Thank you very much for this suggestion. We uploaded more representative fluorescent images. In addition, 15-20 images were taken of myocardial tissue in each group, and the fluorescence intensity was statistically analyzed. These revisions could be found in Figure 1F-G.

Reviewer point #2: Figure 4D: In lane shPEDFR, ZO-1 expression level relative to Tubulin looks lower than ctrl. The author should repeat the western blot to strengthen their conclusion.

Author response #2: Thank you very much for your careful work. Regarding your concern, we increased the number of samples (from n=4 to n=6) and replaced representative immunoblots that better matched the final statistical results. These revisions could be found in Figure 4D-E.

Reviewer point #3: It is clear in figure 5F that, 67LR is necessary for PEDF protective function, while it’s hard to convince me that PEDFR is not involved. Adding cell membrane marker and re-analyzing data will be appreciated.

Author response #3: Thank you for your professional suggestion. Previous studies have found that PEDF can play the biological function of protecting cells through PEDFR (Reference: PMID: 27973457). However, we found that in the OGD environment, PEDF mainly played a protective function of endothelial barrier through 67LR, which was independent of PEDFR. This may be due to different cell types. PEDF is one of the most potent antiangiogenic factors known to date. Interestingly, PEDF usually targets immature new blood vessels while protecting mature vascular systems (Reference: PMID: 35459189). We speculated that the anti-angiogenesis effect of PEDF might affect the function of PEDFR. We will try to solve this problem in the follow-up study. We hope this does not affect the reasonability of the manuscript.

Reviewer point #4: Gao et al. reports that (Gao et al., Int J Mol Med, 2020) inhibiting mTOR signaling is cardioprotective during myocardial infarction in rats. A discussion on how mTOR responses to distinct stimuli and regulates cellular functions via multiple mechanisms would improve the significance of the study.

Author response #4: Thank you very much for your reasonable and scientific advice. We studied this research and cited the background and conclusions into the Section Discussion of our manuscript. These revisions could be found in Line 295-299.

Reviewer point #5: Supplementary figure 2 is not referred correctly in the text.

Author response #5: Thank you very much for your careful work and we apologize for our negligence. We have modified it according to your suggestion. This revision could be found in Line 177.

Special thanks to you for your scientific and valuable comments.

Round 2

Reviewer 2 Report

The authors addressed all of my concerns. This paper will be a nice addition to the field.

Author Response

Thank you for your comments and we have made the changes you requested.